# A Comprehensive Approach to Neoadjuvant Treatment of Locally Advanced Rectal Cancer

**DOI:** 10.3390/cancers17020330

**Published:** 2025-01-20

**Authors:** Annalice Gandini, Stefania Sciallero, Valentino Martelli, Chiara Pirrone, Silvia Puglisi, Malvina Cremante, Massimiliano Grassi, Valeria Andretta, Giuseppe Fornarini, Francesco Caprioni, Danila Comandini, Annamaria Pessino, Serafina Mammoliti, Alberto Sobrero, Alessandro Pastorino

**Affiliations:** Medical Oncology Unit 1, IRCCS Ospedale Policlinico San Martino, 16132 Genoa, Italy; gandini.annalice@gmail.com (A.G.);

**Keywords:** locally advanced rectal cancer, total neoadjuvant treatment, therapeutic algorithm

## Abstract

Locally advanced rectal cancer (LARC) represents a complex disease for three main reasons: (1) the prognosis is related to multiple clinical-radiological parameters; (2) the range of possible therapeutic options has expanded enormously in recent years both in terms of escalation and de-escalation strategies; (3) the choice of treatment must necessarily consider patients’ preferences as the implications for quality of life are extremely impactful. In this review, we attempt to outline a treatment algorithm considering all these variables.

## 1. Advancements in Rectal Cancer Treatment from the Late 1980s

The management of Locally Advanced Rectal Cancer (LARC) has witnessed gradual progress since the late 1980s, with significant improvements, especially in local recurrence (LR) and overall survival (OS).

These advancements have been based on four milestones:(1)The introduction of Total Mesorectal Excision (TME) in 1986 marked a significant enhancement in local control. Before the TME era, the LR rate for LARC was around 30–40% [1];(2)By the late 1990s, a Swedish [2] and a Dutch study [3] demonstrated that preoperative short-course radiotherapy (SCRT) reduced LR to 3–11%, in contrast to surgery alone (8–27%). SCRT afforded a 10% absolute increase in OS in the Swedish trial. However, 30% of patients in these studies had stage I disease, and TME was not mandatory; therefore, it is not possible to be certain about the extent of the survival benefit provided by SCRT;(3)The EORTC 22921 study by Bosset [4] and the FFCD 9203 study by Gerard [5]. These trials demonstrated that fluorouracil added to preoperative radiotherapy (RT) halved the incidence of LR (approximately 8% vs. 16%);(4)In 2004, the CAO-ARO-AIO 04 trial led by Sauer [6] brought forth the latest advancement in local control. Preoperative chemoradiotherapy (CRT) reduced LR compared to post-operative CRT (6% vs. 13%, *p* = 0.006), with a notable improvement in tolerability. However, there was no discernible impact on Disease-Free Survival (DFS) and OS, confirmed in the 10-year follow-up (FU) analyses [7].

In contrast with these advancements, several setbacks have been encountered. The addition of oxaliplatin to fluorouracil-based CRT yielded several negative trials [8,9,10,11,12]. Also, adjuvant chemotherapy (CT) trials provided disappointing results, with four randomized studies reporting negative results or being prematurely closed for poor accrual [4,13,14,15]. Thus, from 2004 to 2020, the standard approach for LARC treatment has been neoadjuvant CRT followed by TME with or without adjuvant CT, depending on the risk of systemic spread. This strategy has resulted in excellent locoregional control (5% LR rate), but the distant metastases rate remained high (30–35%), and this type of strategy carried a significant burden of toxicity (impotence, incontinence, and permanent colostomy), heavily affecting patients’ quality of life.

All these advancements are applicable to patients with pMMR/MSS LARC since, after the advent of immunotherapy, the clinical management of rectal cancers with a deficit in the Mismatch Repair system or Microsatellite Instability (dMMR/MSI) is undergoing a seismic shift that we will discuss in the dedicated paragraph. However, dMMR/MSI accounted for only 3% of all rectal cancers, and the consequent clinical impact is limited to a very small subpopulation of LARCs [16].

## 2. The Definition of Risk Through Appropriate Staging

Along with the advancements in treatments, crucial developments in staging techniques have also been made. In the pivotal trials of neoadjuvant CRT in LARC patients [4,5], the parameters considered for patient accrual were established through digital rectal examination (DRE), measurement of distance from the anal verge by rigid proctoscopy, and CT scan with or without endorectal ultrasound (EUS), as in the EORTC 22921 trial and in the FFCD trial. While these studies set the stage for the modern treatment of LARC, Magnetic Resonance Imaging (MRI) was not yet implemented as a staging technique.

The paradigm shift for the staging of rectal cancer occurred in 2006 with the MERCURY trial [17], which demonstrated the accuracy of pelvic MRI in predicting the negativity of the circumferential resection margin (CRM) on the histological specimen. Specifically, among 408 consecutive patients with all stages of rectal cancer undergoing MRI before TME, the specificity for predicting a clear CRM by MRI was 92%. This finding became even more relevant over time, as MRI-involved CRM was the only preoperative staging parameter significantly correlated with OS (5-year OS 62.2% for MRI-clear CRM compared to 42.2% in MRI-involved), DFS (5-year DFS 67.2% for MRI-clear CRM compared to 47.3% in MRI-involved), and LR on multivariate analysis (HR 3.5 for MRI-involved CRM) [18]. In addition to CRM, MRI should assess other 12 parameters guiding the decision-making in the neoadjuvant setting [19,20,21,22,23]:

Tumor location (upper, medium, low rectum, anterior, posterior, lateral right or left). Distance from the anal verge (cm).

Position of the tumor related to the peritoneal reflection (intraperitoneal vs. extra-peritoneal).

(a)Depth of rectal and perirectal infiltration (T1, T2, T3 a-b-c-d, T4a-b);(b)Nodal status is defined on the maximum diameter (uncertain > 9 mm, at least two uncertain nodes 5–8 mm, at least three uncertain nodes < 5 mm) and shape of the nodes (smooth, irregular border, heterogeneous);(c)Tumor deposits (TD) (positive, negative);(d)Extramural venous invasion (EMVI) (positive, negative);(e)Presence of mucine;(f)Minimal distance from the primary tumor or mesorectal positive lymph nodes and MesoRectal Fascia (MRF) (mm);(g)Distance from the anorectal junction (cm);(h)Caudo-cranial tumor length (cm);(i)Sphincter infiltration (internal sphincter, intersphincteric plan, external sphincter).

Therefore, since MRI is the most appropriate and efficacious staging technique in identifying prognostic factors of local and distant recurrence, it is widely considered mandatory to properly manage rectal cancer [24,25], and it is recommended by both NCCN (Version 4.2024) and ESMO guidelines [26].

While it is commonly recognized that in routine clinical practice, the selection of neoadjuvant treatment is primarily guided by radiological TNM staging, Lord et al. [27] explored the prognostic impact of EMVI, TD, and CRM in comparison to TNM-based staging (or NICE-staging) in 2022. Patients with positive CRM, TD, or EMVI were defined as MRI-high risk, whereas T3+ and/or T4 were defined as TNM/NICE-high risk. The retrospective analysis of 378 English LARC patients demonstrated that prognosis was better predicted by the status of EMVI, TD, and CRM than by T and N. Consequently, the high-risk patients who most likely benefit from preoperative RT would be better recognized through established MRI prognostic factors (EMVI, TD, CRM) rather than by the TNM/NICE 2020 criteria. As a counterpart, the identification of MRI and implementation of EMVI and TD in clinical practice is extremely difficult due to the high expertise required to describe these parameters on MRI imaging.

## 3. Total Neoadjuvant Treatment: Nuances of the New Standard of Care

In 2021, the intensification of neoadjuvant treatment started to be implemented, defining the Total Neoadjuvant Treatment (TNT) strategy. The results obtained in terms of local control and survival supported the TNT as a new standard of care for LARC. However, the implementation of TNT is not straightforward due to the diverse CT and RT treatment regimens, making a “one-size-fits-all” approach not applicable in daily clinical practice. Instead, it requires thoughtful consideration of the risk of recurrence based on initial MRI staging (TNM parameters, EMVI, CRM, TD), the site at higher risk of recurrence (local or distant), and the treatment goal (surgery versus organ preservation). The main strategies nowadays available are graphically summarized in Figure 1.

TNT can be grouped into two main strategies:
(1)TNT with induction chemotherapy (INCT)
(a)Chemo doublet (CAPOX/FOLFOX) followed by CRT [28,29,30,31];(b)Chemo triplet (FOLFIRINOX) followed by CRT [32,33].(2)TNT with consolidation chemotherapy (CNCT)
(a)SCRT followed by chemo doublet (CAPOX/FOLFOX) [34,35,36,37];(b)CRT followed by chemo doublet (CAPOX/FOLFOX) [29,30,31,38,39].

### 3.1. TNT with Induction Chemotherapy

In this schedule of treatment, CT is administered before CRT.

(a)Chemo doublet followed by CRT.

The first prospective randomized trial evaluating this strategy was conducted in 2015 by Fernandez-Martos and colleagues, who randomized 108 LARC patients (T3–T4 or N+) to receive four cycles of CAPOX followed by CRT and surgery versus CRT followed by surgery and four cycles of adjuvant doublet CT. No differences in survival were found even if patients’ characteristics were not well balanced between the two arms, with a higher rate of T4 and G3 tumors in the induction arm. However, INCT was characterized by lower toxicity and improved compliance compared to adjuvant treatment [28].

The CAO-ARO-AIO-12 study randomized 302 LARC patients to receive three cycles of induction FOLFOX followed by CRT versus the inverse sequence of CRT followed by three FOLFOX. In both arms, patients underwent radical surgery by TME because NOM was not an option. INCT produced a lower pathological Complete Response (pCR) (pCR rate 17% vs. 52%, *p* < 0.001) compared to CNCT. No differences in DFS (3-year DFS 73%) and OS were found [29].

The phase II OPRA trial randomized 324 patients with stage II or III to INCT-CRT versus CRT-CNCT followed by TME or watch-and-wait strategy (WW) for patients achieving clinical complete response (cCR) or near-cCR. CT delivered in the induction and consolidation phase was FOLFOX (eight cycles) or CAPOX (five cycles). The 3-year DFS was 76% in both arms. After a median FU of 5.1 years, no difference in 5-year DFS was observed between treatment arms (71% vs. 69% *p =* 0.675) [30,31].

(b)Chemo triplet followed by CRT.

The phase III PRODIGE-23 study randomized 461 LARC (cT3–cT4) to INCT with six cycles of FOLFIRINOX, followed by CRT, surgery, and 3 months of adjuvant CT versus CRT followed by surgery and 4 months of adjuvant doublet CT. With a median follow-up of 7 years, the INCT arm demonstrated a DFS gain of 5.1%, an OS gain of 5.8%, and a metastasis-free survival (dMFS) gain of 6.9% when compared to standard treatment [33]. The maturity of follow-up, along with the consistency of the incremental gains among the endpoints, makes FOLFIRINOX the most efficacious scheme of INCT. The compliance to INCT with triplet was very high despite the worse toxicity profile [32].

*Interpretation and takeaways*: INCT represents a valid option for patients with LARC. The reasons supporting this strategy in clinical practice are the efficacy in early eradication of micrometastases, rapid clinical benefit on symptomatic tumors, and reduction of the target volume for RT. The choice between doublet (CAPOX/FOLFOX) or triplet (FOLFIRINOX) is based on the patient’s condition and age. However, it should be noted that the only induction trial that significantly impacts survival is the PRODIGE-23 with the triplet. Thus, when possible, induction of FOLFIRINOX for six cycles should be the preferred regimen. Alternatively, 3–4 months of CAPOX/FOLFOX are acceptable. Of course, early T3 a/b and N0 without other risk factors should be excluded because of the risk of overtreatment.

Table 1 reports the results of the main trials of INCT, as reported above.

### 3.2. TNT with Consolidation Chemotherapy

Another established TNT strategy is the delivery of CT as a consolidation treatment between CRT/SCRT and surgery.

(a)SCRT followed by chemo doublet.

The first trial to explore the consolidation strategy was the Polish-II study that randomized 515 patients with fixed cT3 or resectable cT4 rectal cancers to SCRT followed by three cycles of FOLFOX and surgery versus CRT, surgery, and adjuvant CT. No difference in R0 resection, OS, and DFS were found, with a similar toxicity profile compared to the control arm [34]. In this trial, patients with more advanced T stage were enrolled as compared with the other studies (64% cT4 and 55% low-lying).

The RAPIDO trial [35] randomized 920 high-risk LARC patients (30% cT4, 30% EMVI, 65% cN2, 60% involvement of the mesorectal fascia and 15% enlarged lateral lymph nodes). The experimental arm was SCRT followed by CNCT with six cycles of CAPOX or nine cycles of FOLFOX and surgery compared to CRT followed by TME and optional adjuvant treatment. At 3 years, the distant metastasis rate (dMR) was significantly reduced in the experimental arm (20 vs. 26.8%); on the other hand, LR was almost two-fold higher in the experimental arm (10 vs. 6%) at 5 years [36]. This can suggest that the intensification of the CT treatment (INCT or CNCT) reduces systemic failures; however, SCRT is not the optimal treatment option in LARC at high risk of local relapse as those enrolled in the RAPIDO.

The STELLAR study by Jin et al. [37] enrolled 599 patients who were randomly assigned to SCRT followed by four cycles of CAPOX before TME and two more cycles after surgery versus CRT followed by TME and subsequent adjuvant CT with six cycles of CAPOX. Most of the patients were T3 (82–85%) and within 10 cm from the anal verge (98–100%). Almost 50% had EMVI and/or MRF positive. Both cCR and pCR (cCR 11.1% vs. 4.4%; pCR 21.8% vs. 12.3%) were higher in the TNT arm. LR rate was lower in the experimental arm when compared to the standard am (8.4% vs. 11%). The 3-year DFS rate in this study was slightly lower than that in other trials (see Table 2), and no difference was seen in the distant metastasis rate among the two arms. Similar to other results, the compliance with the treatment was good even if a higher rate of toxicity G3-4 was reported with the TNT regimen (26.5 vs. 12.6%).

(b)CRT followed by chemo doublet.

In 2011, Garcia-Aguilar and colleagues conducted the TIMING trial, a non-randomized phase II study, in which 144 patients with clinical response after CRT received 2, 4, or 6 additional cycles of FOLFOX before surgery. Timing to surgery was 4 weeks after CRT or 3–5 weeks after the last cycle of CNCT, reaching an average of 11 weeks. pCR was higher in the experimental group (25% vs. 18%). However, it is known that the timing between RT and surgery correlates with the pCR rate; therefore, it is not possible to quantify the real contribution of CNCT [38,39].

As already explained, the CAO.ARO.AIO-12 and the OPRA trial were designed with an INCT and a CNCT arm. Survival rates were similar in both studies. The CAO.ARO.AIO-12, CNCT produced a higher rate of pCR (52% vs. 17%, *p* < 0.001), which was not associated with an increase in survival [29]. In the OPRA trial, an increase in TME-free survival (54% vs. 39% *p* = 0.012) was observed with the consolidation strategy. Indeed, although no difference in survival was found, a significant difference in the number of patients who could preserve the rectum was shown: 39% in the INCT arm vs. 54% in the CNCT arm, emphasizing the relevance of patient’s surgical preference in decision making [30,31].

*Interpretation and takeaways*: CNCT can be used both after CRT and SCRT. The evidence leading to a reduction in distant metastases and a consequent increase in DFS is linked to the RAPIDO trial. However, the RAPIDO strategy is burdened by a significant increase in LR, making the SCRT regimen followed by consolidation strategy contraindicated in high-risk patients (T4, CRM+, EMVI+, positive lateral lymph nodes), as stated in Section 3.2, a. The consolidation strategy after CRT has not demonstrated an impact on DFS but seems to ensure good local disease control due to long-course RT and intuitively allows good control over micrometastases thanks to 3–4 months of doublet CT.

The results of the main trials of CNCT are reported in Table 2.

*TNT induction vs. consolidation*. The studies comparing doublet INCT versus CNCT (with CRT as the RT technique) are phase II trials that failed to demonstrate an increase in survival in favor of one or the other strategy. The increase in pCR is afforded by consolidation in the CAO.ARO.AIO-12 trial can be related to a longer interval between CRT and surgery. The most common interpretation of the results of the OPRA study is that when the goal of the strategy is to achieve cCR and NOM, then CRT followed by CAPOX/FOLFOX consolidation appears to be the best strategy (+15% TME-free survival at 5 years). However, we should consider that even with INCT followed by CRT, the 5-year TME-free survival was 39%, and the pCR rate was 28% with high efficacy induction regimens such as in PRODIGE-23, leaving room to consider NOM management even after the induction strategy.

## 4. NOM—Precise Quantification of Risk Is Mandatory to Implement the Strategy

The major prospective randomized evidence on the value of the NOM strategy is derived from the OPRA trial [30,31]. After TNT, tumor restaging was performed by DRE, endoscopic examination, MRI, and total body CT scan within 8 weeks; biopsy was not required. Patients who achieved a cCR or near cCR were offered WW, which required a strict surveillance protocol. Regrowth was described in 40% of patients in INCT and 27.5% in CNCT. At a median FU of 56 months, 94% of LR occurred within 2 years and 99% within 3 years. Of particular interest is the rate of TME-free survival: this was 54% at 5 years in the CRT-CNCT arm versus 39% in the INCT-CRT arm (delta of 15%). However, the 5-year DFS after TME at restaging or at the tumor regrowth was comparable (61% vs. 62%).

Recently, at the European Society of Medical Oncology (ESMO) Congress 2024, the first results of the Italian phase II NO-CUT trial were presented. One hundred and eighty patients with mid/low cT3-4 and/or cN1-2, pMMR/MSS rectal adenocarcinoma were included and treated with INCT (four cycles of CAPOX) followed by CRT. NOM was offered to patients with cCR. After a median FU of 27 months, cCR was achieved in 26% of patients, and these proceeded to NOM, with the lower cT stage confirmed to be a clinical predictor of cCR. Distant relapse-free survival (DRFS) at 30 months was 77% in the overall population and 97% in the NOM population. The organ preservation rate was 85% [40].

Other indirect data on NOM were derived from the long-term FU of the CAO/ARO/AIO-12 trial, in which 10 patients with cCR chose the NOM, even if it was not a protocol option. Specifically, 8 out of 10 of these patients are still sustaining cCR [41].

A non-randomized trial has evaluated the safety of NOM in 86 patients with stage I–III rectal adenocarcinoma after SCRT and CNCT (8–12 cycles of FOLFOX or CAPOX): 76% were stage III and 50% had low rectum tumors. Clinical response was assessed by digital RE, pelvic MRI, and endoscopy: 50% of the patients obtained a clinical response, with an association with less advanced T and N stage and less CRM involvement (23% vs. 65%). The 2-year LR rate was 21% in all the patients; all of them successfully underwent salvage surgery, and none of them relapsed yet. The 2-year TME-free survival was 40%, reaching 69% in the patients who had cCR. At the time of the data analysis (median FU 30 months), no distant recurrences occurred [42].

From an analysis of the International Watch & Wait Database, the dMR appeared to be significantly higher in patients who had an LR (18% vs. 8%), with a decrease in 5-year disease-specific survival (DSS) of 10% (84% vs. 94%) [43]. Similar results were obtained from a retrospective analysis from the Memorial Sloan Kettering, in which 71% of the patients received TNT. The dMR was significantly higher in patients with local regrowth (36%) when compared with patients with a sustained cCR (1%) [44]. A retrospective Spanish study showed a dMR of 6.4% in patients with pCR at surgery (5-year OS 89.3%) [45]. At the ASCO GI Congress 2024, Fernandez et al. demonstrated that local regrowth after WW strategy is an independent risk factor for the development of distant metastases (22.8 vs. 10.2%), even if these patients did not receive a TNT regimen and data were biased by the retrospective design of the analysis [46]

Another line of research, within the framework of the NOM, regards the possibility of performing the endoscopic local excision (LE) of the residual tumor after neoadjuvant therapy. The GRECCAR-2 trial [47] randomized 148 patients with early rectal tumors (cT2–T3) in clinical response after CRT to TME versus LE. At 5 years FU, the LE strategy showed non-inferiority to standard surgery in terms of local and distant recurrence. However, when salvage TME or Miles surgery is indicated due to LR, patients treated with LE had a higher risk of surgical complications and post-operative dysfunctions. Moreover, the role of TNT strategy and LE still needs to be defined, and the ongoing GRECCAR-12 study is likely to address this issue. Hence, the LE strategy could be considered only in centers with advanced expertise and for patients who have not achieved cCR and are willing to avoid TME resection.

*Interpretation and takeaways*. The use of the TNT strategy substantially increased the likelihood of achieving a cCR, which is around 30% of patients with LARC. The option of NOM implies the need to evaluate the cost–benefit ratio compared to the standard option of surgical intervention.

On the one hand, the standard approach of surgical resection for LARC in cCR is supported by robust randomized data, resulting in LR rates <5% and 3-year DFS exceeding 75% in pivotal trials. However, these robust results are burdened by severe long-term quality-of-life sequelae and potential permanent colostomy.

On the other hand, retrospective data agree that the risk of local regrowth of LARC in cCR was initiated on NOM at 30% within 3 years of FU [43]. However, local regrowth is salvageable with radical surgical in over 90% of cases.

Another potential pitfall of NOM in cCR is that historical data suggest an increased risk of metastases. At the same time, the OPRA and, even more, the NO-CUT results are reassuring, with a 30-month dMFS of 97% in NOM patients with sustained cCR.

At least two points on the clinical management in WW still need to be clarified. The first open question is how to define cCR. Evidence seems to suggest that, when NOM is the goal, endoscopic reassessment should be associated with MRI imaging, digital rectal examination, and CEA level.

The second issue is the optimal timing and structure of FU timing. It is important to define a precise FU in order to spare surgery to responders but, at the same time, be able to promptly identify a local regrowth. It is known that in responder patients, the “late” disease reassessment at 12 weeks after TNT is associated with higher rates of pCR [48,49]. As a counterpart, a late reassessment of disease among non-responder patients correlates with reduced survival [50].

It is important to remember that the incidence of colorectal cancer, and in particular rectal cancer, is increasing among young patients, defined as those less than 50 years old. Due to their longer life expectancy and the impact that a mutilating rectal surgery can have on their quality of life, it is mandatory to better understand the long-term oncological risk of NOM in this subpopulation. The prognosis of early-onset colorectal cancer is still debated [51,52], but a multicenter analysis suggests that the outcome of NOM after cCR is comparable with patients older than 50 years [53]. Shared decision-making between the patient and the physician will determine the adoption of one strategy over the others, or rather, choosing between the certainty of low recurrence risk, burdened by high side effects, and a less invasive approach that, however, entails a significant risk of local regrowth.

### The Management of the Follow-Up

Unlike other cancer types, where follow-up protocols have long been standardized, locally advanced rectal cancer requires a personalized FU plan based on the chosen treatment approach. While no significant changes have been observed in the management of patients undergoing TNT followed by surgery [26], the NOM option needs a carefully tailored FU strategy to ensure the early detection and treatment of LR, given the high likelihood of achieving a cure. As we have already discussed, around 1 out of 3 patients developed an LR after NOM; nevertheless, long-term outcomes seem to be preserved if salvage surgery is promptly performed [31]; it follows that timely identification of an LR is mandatory. Furthermore, LR may anticipate distant recurrences, although this risk cannot be precisely determined [46]. Consequently, regular systemic surveillance is mandatory. Unfortunately, due to the shades of uncertainty and the lack of randomized trials, there is no worldwide consensus on the best FU approach after NOM is reached. Among the available data, most LRs occur within the first two years after the completion of treatment. Therefore, there is a general agreement to intensify FU in the first 2–3 years [54]. National and international guidelines agree on the need to reassess patients locally every 3 months through clinical examination, digital rectal examination, and measurement of the tumor marker CEA. For pelvic MRI, the recommended interval ranges from 3 to 6 months, while reassessment with CT scans should be performed every 6 months. Table 3 presents and compares the most frequently adopted FU strategies to date. Patients who are offered the W&W strategy should be fully aware of the necessity of complete adherence to such a strict FU program, and clinicians should be able to ensure the availability of timely exams and raise concerns about NOM in non-compliant patients. Surely, a better understanding of the disease and the identification of prognostic factors allowing patients’ stratification is highly needed, and both clinical and translational biomarkers are awaited.

## 5. Radiotherapy Omission

Evidence supporting the omission of RT from neoadjuvant treatment has emerged in 2023, and this new option further challenges the standards of LARC treatment.

The most relevant study in this regard is the PROSPECT trial, which randomized over a thousand low-risk LARC patients (no T4, no N2, no candidates for Miles surgery, CRM > 3 mm on MRI) to receive six cycles of induction FOLFOX and, in case of tumor reduction > 20% assessed by MRI and endoscopy, surgery was performed, followed by other six cycles of adjuvant FOLFOX. If the tumor did not achieve sufficient shrinkage, patients underwent CRT followed by subsequent surgery and adjuvant CT. CRT, followed by surgery and adjuvant therapy, was the control arm. The study demonstrated the non-inferiority of selectively omitting RT in 5-year DFS and local control [57].

The CONVERT trial is a randomized phase III non-inferiority study with a design very similar to the PROSPECT study. Almost 600 LARC patients (including T4, low-lying cancers but no CRM positive) were allocated to 4 cycles of CAPOX followed by surgery and four cycles of CAPOX as adjuvant therapy versus CRT followed by surgery and adjuvant therapy. The primary endpoint of 3-year LR-free survival was not statistically met, but the crude rate of local relapses (3%), as well as 3-year DFS and OS, were similar in both arms [58].

Another indirect evidence supporting the omission of RT comes from the Chinese FOWARC trial. This study randomized LARC patients (including T4 and N2) to three arms of treatment: CRT followed by surgery and adjuvant CT, CRT intensified with concomitant oxaliplatin followed by surgery and adjuvant CT or 4–6 cycles of FOLFOX followed by surgery and 4–6 cycles of FOLFOX as adjuvant (no RT arm). Consistently with the results of PROSPECT and CONVERT, this study did not show an increase of LR in the “RT-free” arm [59].

*Interpretation and takeaways*: Analyzing the overall data from these studies, the rate of LR is extremely low even in the absence of RT, ranging from 2% in the PROSPECT [57] to 9% in the FOWARC [59] trials. However, it should be emphasized that the selective omission of RT is currently reserved for low-risk patients (no cT4, no N2, no EMVI, negative CRM, not candidates for Miles surgery) who achieve tumor downstaging after six cycles of induction FOLFOX. In these patients, the use of RT might be considered an overtreatment, negatively impacting the quality of life and providing no advantage in terms of local control [60].

## 6. dMMR: The Immunoablative Treatment

Approximately 3% of rectal cancers are dMMR/MSI [16]. Despite the low frequency, it is mandatory to identify these patients at the time of diagnosis since dMMR/MSI is strictly related to outstanding response to immunotherapy, resistance to CT, and genetic predisposition (Lynch Syndrome). Indeed, around 70–80% of dMMR/MSI rectal tumors can identify a hereditary cause related to Lynch Syndrome [61,62]. The identification of these patients allows their affected relatives to enroll in surveillance programs, reducing mortality by 60% [63].

The proof of principle about the response to immunotherapy came at ASCO 2022 when Dr. A. Cercek presented for the first time the results of neoadjuvant dostarlimab administered for nine cycles to 12 patients with dMMR/MSI LARC (T3–T4 N0 or any T, N+), that revolutionized the treatment in these patients [64]. All patients showed cCR, which is defined as no clinical, radiological, or endoscopic evidence of a tumor. After this amazing response, patients were all treated with NOM, and none of them received CRT or surgery, significantly reducing severe and impactful consequences on quality of life [16]. After a median FU of 28.9 months at the last update in ASCO 2024, no progression or recurrence was reported in any of the 41 patients who completed the treatment [65]. Despite these astonishing results, dostarlimab is not approved anywhere in the world for the frontline treatment of locally advanced dMMR rectal cancer, and a longer FU is needed to monitor the time to recurrence or progression.

Two other anti-PD1 single agents have been evaluated in the same setting.

Chen et al. designed a Chinese, open-label, single-arm, phase 2 study in which 16 patients with dMMR/MSI LARC received four cycles of neoadjuvant sintilimab followed by surgery and adjuvant sintilimab with or without CT or eight cycles of neoadjuvant sintilimab followed by surgery or NOM (only for patients with cCR). Overall, 15 patients had a response, with 12 cCR, and only one patient had progressive disease. Ten patients chose NOM, while six underwent surgery; among these, pCR was 50%. No new safety signals were reported [66].

Neoadjuvant pembrolizumab was also evaluated in a series of patients with dMMR/MSI solid tumors in a phase II trial conducted by Ludford et al.: among the 35 patients enrolled, 8 had LARC. Unlike Cercek’s patients, 2/8 showed progressive disease; however, the remaining 6 (75%) obtained a clinical response, and pCR was reported in the only patient who underwent surgery [67].

*Interpretation and takeaways*: Ablative immunotherapy is the most impactful advancement ever in LARC patients, who must be tested for dMMR/MSI at the time of diagnosis. Dostarlimab could become the best option in this setting, but for the moment, it has not been approved, and more FU is awaited. Potential concerns regarding the type of surveillance after NOM, as well as the high disparity reported between radiological and pathological outcomes, make a radiographic definition of cCR challenging. Moreover, there is a high diversity among trials in terms of the duration of the therapy, which remains controversial, ranging from 3 to 6 months with or without an adjuvant phase, and whether a double immunotherapy is necessary or not. Therefore, caution is still needed, and more data are required, but this approach is likely going to revolutionize the management and the prognosis of these patients, who can finally benefit from a more effective and less toxic treatment.

## 7. The Potential Role of Liquid Biopsy

As extensively detailed above, there is an urgent need for novel biomarkers to improve the clinical outcomes in LARC. One of the most promising methods under investigation is liquid biopsy [68], namely the detection and analysis of any tumor-derived component (e.g., DNA, mRNA, exosomes) in body fluids (e.g., blood, saliva, cerebrospinal fluid, ascites) [69]. Among the various techniques, circulating tumor DNA (ctDNA) in peripheral blood has been the most actively studied in recent years. Its potential uses in clinics may cover a wide range of scenarios, both in earlier and advanced settings [70]. Focusing on LARC, there are four main domains where ctDNA may play a fundamental role [71]:*Prognostic and predictive biomarker at diagnosis*: even though this may be extremely relevant for stratifying the patients, for the moment, evidence has not demonstrated a close relationship between ctDNA positivity at baseline, survival rates, or pCR in locally advanced disease so far [72,73];*Escalation and de-escalation of neoadjuvant therapies*: since ctDNA quantity can be considered a surrogate for disease burden [74], assessing early quantitative variations after the beginning of neoadjuvant treatment may help clinicians to escalate or de-escalate the ongoing therapies;*Treatment selection after surgery*: currently, there is no consensus on the right strategy to adopt after surgery in LARC patients [26,75]. Detecting ctDNA after radical treatment might support the use of adjuvant CT since the RFS in ctDNA-positive patients after surgery is significantly lower than in the ctDNA-negative group [40,72]. Therefore, liquid biopsy can identify patients who might benefit from adjuvant treatment, avoiding unnecessary and toxic therapies for others;*Assessing Minimal Residual Disease (MRD) during FU*: during surveillance, ctDNA could identify disease relapses earlier than conventional radiological assessments [70], which would increase the possibility of curative treatments and, consequently, survival [76].

While the idea of implementing liquid biopsy in clinical settings may seem promising, numerous uncertainties surround its introduction in clinical practice. Firstly, each assay showed a distinct limit of detection and quantitation, influencing the amount of plasma DNA and ctDNA fraction required for informative results, especially in localized disease. Secondly, the optimal timing for sample collection still needs to be demonstrated among the different therapeutic strategies, in which the availability of results may not align with the decision-making timeline [71]. Lastly, existing evidence originates from observational studies or clinical trials with small sample sizes. Therefore, there is an urgent need for robust data from large randomized clinical trials to definitively establish the role of ctDNA in LARC [77]. The AGITG DYNAMIC-RECTAL study represents the first effort in this direction [78] that randomized 230 cT3-4 and/or N+ patients to standard management or ctDNA-informed management arm. The study showed that only 46% of patients in the ctDNA-informed group received CT, compared to 77% in the standard group. The 3-year RFS rates were 82% and 74%, respectively, but unfortunately, the study did not reach the non-inferiority margin of at most 10%, given the early accrual discontinuation due to COVID-19 pandemics and the TNT strategy implementation.

Despite these limitations, the trial offered new insights into the role of ctDNA not only as a biomarker to guide adjuvant CT but also as a prognostic factor, paving the way for the use of ctDNA in clinical trials [78].

## 8. Discussion: A Comprehensive Approach to LARC Tailored on Risk of Relapse, Treatment Goals, and Patients’ Attitude

The landscape of LARC treatment has expanded significantly in the past 5 years, with several therapy options both in terms of de-escalation (omission of RT and surgery) and escalation (total neoadjuvant approach). These two paths of development, although seemingly diametrically opposed, are closely interconnected because TNT increases the chances of omitting RT in case of good response to INCT and sparing of surgery in the case of cCR. In addition to a continuously enriched array of options, the significant impact on the quality of life of each treatment, especially RT and surgery, makes it essential to integrate clinical–radiological parameters with the treatment goal (surgery versus NOM), which stems from a shared decision with the patient. A practical, evidence-based algorithm that might guide clinicians in their daily practice is represented in Figure 2. Options are reported in order of preference based on the opinion of the authors after balancing the pros and cons of different approaches, as reported in the previous sections of this paper.

*Low-risk LARC*. This group of patients involves cT1–T2 and N1 tumors, early cT3 a/b and N0 with no other risk factors (EMVI-, CRM-, TD-, lateral pelvic N negative) of the middle/upper rectum (above 5 cm from the anal verge). In this risk class, if the goal is surgical resection, treatment options range from neoadjuvant CT alone with FOLFOX followed by TME and adjuvant FOLFOX (PROSPECT trial) to upfront surgery followed by adjuvant CT as indicated by the ESMO guidelines [26], and to neoadjuvant CRT followed by TME. However, if the patient strongly leans towards surgery omission, more intensive treatment options can be considered, such as neoadjuvant CRT (STAR TREC [79] and TAU-TEM [80] studies) and CRT followed by CNCT.

*Intermediate-risk LARC*. Patients with late T3 c/d and N+ tumors without other risk factors, such as EMVI, CRM, TD, and lateral pelvic N, are suitable for TNT when the goal is both surgery and cCR. More specifically, when surgery is the goal, the induction triplet with FOLFIRINOX ranks first among the TNT regimens; other options include INCT with doublet or CNCT after CRT/SCRT. When NOM is the objective, CRT followed by CNCT (FOLFOX/CAPOX) is the preferred option given the 54% TME-free survival at 5 years afforded by the OPRA trial. The PRODIGE-23 scheme and the RAPIDO strategy can be considered second and third options, given the very high rates of pCR reported in these trials. Also, the CAPOX/FOLFOX INCT followed by CRT is an option given the 39% TME-free survival at 5 years afforded by this strategy in the OPRA trial.

*High-risk LARC*. Patients with cT4 and/or with other major risk factors (EMVI, CRM, TD, lateral pelvic N), irrespective of the tumor location, deserve the most intensive treatment in order to maximize local control and reduce the risk of metastatic spread. INCT with FOLFIRINOX followed by CRT is the preferred option, and, as an alternative, CRT preceded or followed by CAPOX/FOLFOX. The SCRT followed by CNCT doublet is not indicated in this specific stage, given the increased risk of local relapses (+4%). NOM is generally not recommended in these cases also because the cCR is unlikely and the risk of local and regional regrowth/relapse is >50% [43].

## 9. Conclusions

To conclude, the therapeutic landscape of LARC is becoming increasingly complex following the introduction of TNT, along with the possibilities of escalation and de-escalation treatments. These strategies aimed at maximizing local and distant control while also preserving patients’ quality of life. Therefore, the ideal and comprehensive LARC management should be based on both standard and “new” clinical–radiological parameters that allow for accurate risk assessment, as well as consideration of patient preferences. All these evaluations must be enriched and implemented during the multidisciplinary discussion, which must be considered indispensable. However, the large volume of data from recent trials has made outlining a precise treatment algorithm to guide clinicians in daily practice even more challenging and almost prohibitive. Hence, the therapeutic strategy should be the result of a full-shared decision between doctors and each patient. The definition and implementation of biomarkers remain an unmet need in this field. ctDNA may potentially enter clinical practice as a guide for enhancing neoadjuvant therapy or surgery in the case of persistent positivity or to open the possibility of less intensive approaches in the case of negativization.

## Figures and Tables

**Figure 1 cancers-17-00330-f001:**
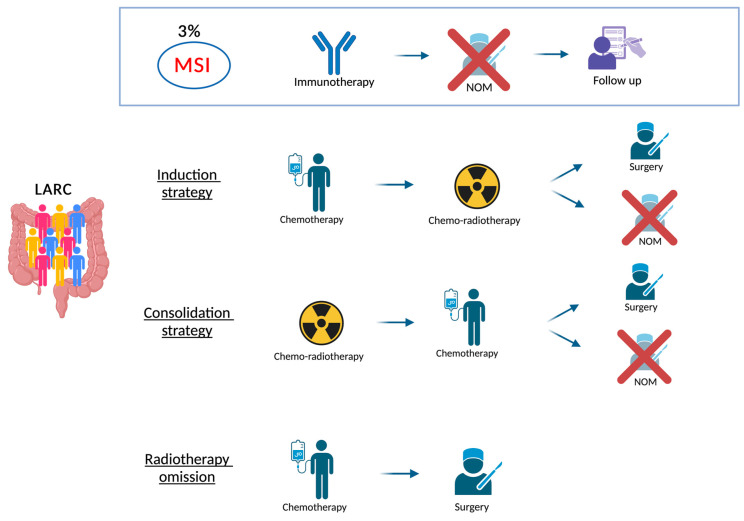
Therapeutic options for locally advanced rectal cancer. Picture created with Biorender.com. Acronyms: LARC = Locally Advanced Rectal Cancer; MSI = microsatellite instability; NOM = Non-operative Management.

**Figure 2 cancers-17-00330-f002:**
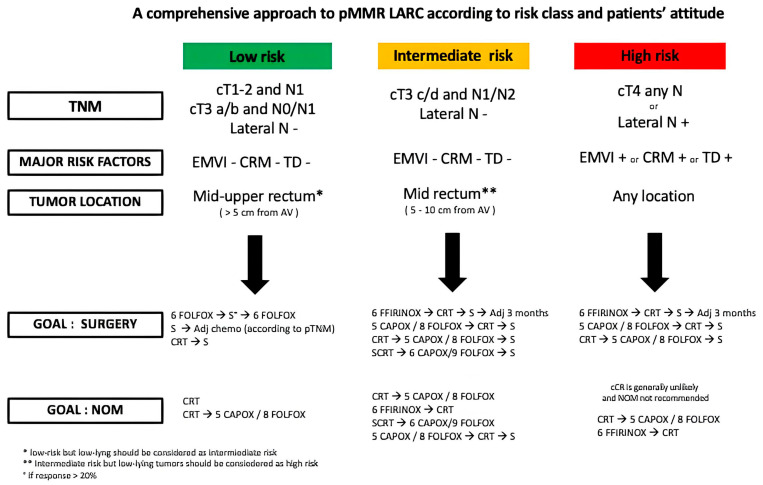
Treatment algorithm for locally advanced rectal cancer.

**Table 1 cancers-17-00330-t001:** Comparison of the main TNT trials with induction chemotherapy.

	PRODIGE-23 [32,33]	OPRA [30,31] Induction Arm	GCR-3 [28]	CAO.ARO.AIO-12 [29] Induction Arm
Phase	III	II	II	II
TNT type	INCT	INCT	INCT	INCT
Neoadj CT regimen	6xFOLFIRINOX	8xFOLFOX/5xCAPOX	4xCAPOX	3xFOLFOX
RT type	LCRT	LCRT	LCRT	LCRT
Control arm	LCRT–TME–CT	LCRT–3xFOLFOX–TME	LCRT–TME– 4xCAPOX	LCRT–3xFOLFOX–TME
Pts charact.	≤15 cm from AV cT3highrisk cT4	Not specified cT3-4 N0 N1-2	≤12 cm from AV T3-4 any N	≤12 cm from AV cT3 low/>cT3b med/cT4/N+
Primary endpoint	3y DFS	DFS	pCR	pCR
Surgery performed	92%	37%	88%	96%
ypCR	27.8%	NA (NOM)	14%	17%
3y LR	4.8%	NA	4%	6%
3y DFS	76%	76%	70%	73%
3y dMFS	79%	82%	82%	82%
3y OS	91%	NA	81%	92%
TRAEs G3-4	46%	NA	NA	15.4%

Acronyms: INCT = induction chemotherapy; LCRT = long course radiotherapy; TME = total mesorectal excision; AV = anal verge; pCR = pathological complete response; DFS = disease-free survival; LR = local relapse; dMFS = distant metastasis-free survival; OS = overall survival; TRAEs = treatment-related adverse events; NA = not available.

**Table 2 cancers-17-00330-t002:** Comparison of the main TNT trials with consolidation chemotherapy.

	OPRA [30,31] Consolidation Arm	TIMING [39]	RAPIDO [35]	STELLAR [37]	CAO.ARO.AIO-12 [29] Consolidation Arm	POLISH-II [34] Consolidation Arm
Phase	II	II	III	III	II	III
TNT type	CNCT	CNCT	CNCT	CNCT	CNCT	CNCT
Neoadj CT regimen	8xFOLFOX/5xCAPOX	FOLFOX 2-4-6 cycles	9xFOLFOX/6xCAPOX	4xCAPOX	3xFOLFOX	3xFOLFOX
RT type	LCRT	LCRT	SCRT	SCRT	LCRT	SCRT
Control arm	LCRT–5xCAPOX/8xFOLFOX-TME	No control arm	LCRT–6xCAPOX/9xFOLFOX–TME	LCRT–TME–6xCAPOX	3xFOLFOX–LCRT–TME	LCRT–TME–Adjuvant
Pts charact.	Not specified cT3-4 N0 N1-2	≤12 cm from AV cT3-4, N0 N1-2	≤16 cm from AV cT4a-b cN2 EMVI+ MRF involv. Lateral N+	≤10 cm from AV cT3-4 N+	≤12 cm cT3 low >cT3b med cT4 N+	cT3 fixed cT4
Endpoint	DFS	pCR rate	DRTF	3y DFS	pCR	R0 resection
Surgery performed	49%	96%	92%	77.8%	97%	NA
ypCR	NA (NOM)	25%	28.4%	16.6%	25%	16%
3y LR	NA	NA	8.3%	8.5%	5%	22%
3y DFS	76%	NA	76.3%	64.5%	73%	53%
3y dMFS	84%	NA	80%	77.1%	84%	NA
3y OS	NA	NA	89.1%	86.5%	92%	73%
TRAEs G3-4	NA	16%	48%	26.5%	17.4%	24%

Acronyms: CNCT = consolidation chemotherapy; LCRT = long-course radiotherapy; SCRT = short-course radiotherapy; TME = total mesorectal excision; AV = anal verge; pCR = pathological complete response; DFS = disease-free survival; LR = local relapse; dMFS = distant metastasis-free survival; OS = overall survival; TRAEs = treatment-related adverse events; NA = not available.

**Table 3 cancers-17-00330-t003:** Surveillance programs according to NCCN guidelines (Version 4.2024), the Dutch Watch-and-Wait Consortium [55] and guidelines, and the Brazilians indications [56]. Acronyms: CE = clinical examination; RE = rectal examination; CEA = carcinoembryonic antigen; Proct = proctoscopy; CEA = carcinoembryonic antigen; CT = computed tomography; MRI = Magnetic Resonance Imaging; QxM = every x month.

Exams	1st Year			2nd Year			3rd Year			4th–5th Year		
	NCCN	Dutch	Brazil	NCCN	Dutch	Brazil	NCCN	Dutch	Brazil	NCCN	Dutch	Brazil
CE; RE	Q3M	Q3M	Q2M	Q3M	Q6M	Q2M	Q6M	Q6–12M	Q2M	Q6M	Q6–12M	Q6M
CEA	Q3–6M	/	/	Q3–6M	/	/	Q6M	/	/	Q6M	/	/
Proct.	Q3M	Q3M	Q2M	Q3M	Q6M	Q2M	Q6M	Q6–12M	Q2M	Q6M	Q6–12M	Q6M
MRI	Q6M	Q3M	Q3M	Q6M	Q6M	Q3M	Q6M	Q6–12M	Q3M	/	Q6–12M	Q6M
CT scan	Q6–12M	?	Q6M	Q6–12M	?	Q6M	Q6–12M	Q12M	Q6M	Q6–12M	Q12M	Q6M

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
