# Peer review of "A Comprehensive Approach to Neoadjuvant Treatment of Locally Advanced Rectal Cancer"

_cancers, 2025, doi:10.3390/cancers17020330_

Round 1
Reviewer 1 Report
Comments and Suggestions for Authors
The authors extensively reviewed the literature and reported a comprehensive summary on a hot topic, perfectly highlighting the most innovative points such as TNT and NOM.
I would only add a comment on the role ( if any) and the possible relationship btween endoscopic resection and CT.
Author Response
1) The authors extensively reviewed the literature and reported a comprehensive summary on a hot topic, perfectly highlighting the most innovative points such as TNT and NOM. I would only add a comment on the role (if any) and the possible relationship between endoscopic resection and CT.
1) We would like to thank the reviewer for the time spent in reading our paper and for the appreciation shown. In the paragraph regarding NOM, we have added a comment on local excision, as follows:
“Another line of research, within the framework of the NOM, regards the possibility to perform the endoscopic local excision (LE) of the residual tumor after neoadjuvant therapy. The GRECCAR-2 trial (48), randomized 148 patients with early rectal tumors (cT2-T3) in clinical response after CRT, to TME versus LE. At 5-years FU, the LE strategy showed non-inferiority to standard surgery in terms of local and distant recurrence. However, when salvage TME or Miles surgery is indicated due to LR, patients treated with LE had a higher risk of surgical complications and post-operative disfunctions. Moreover, the role of TNT strategy and LE still needs to be defined, and the ongoing GRECCAR-12 study is likely to address this issue. Hence, the LE strategy could be considered only in centers with advanced expertise and for patients who have not achieved cCR and are willing to avoid TME resection.”
Reviewer 2 Report
Comments and Suggestions for Authors
The paper is basically a broad review of very well-known topics. In any case, it is useful to those who approach the subject, in particular it guides surgeons to always consider a neoadjuvant phase. Discussion should be divided into discussion and Conclusion. Conclusions are missing here. Chapter 8 is not scientifically clear and appears too long and should be anticipated before discussion and conclusion. Talking about the usefulness or otherwise of the liquid biopsy does not seem necessary, it is not essential for the topic treated. Maybe you propose it in another article.
Why Interpretation and takeaway is underlined? take off pls.
Author Response
Reviewer 2:
1) The paper is basically a broad review of very well-known topics. In any case, it is useful to those who approach the subject, in particular it guides surgeons to always consider a neoadjuvant phase. Discussion should be divided into discussion and Conclusion. Conclusions are missing here.
1) We thank the reviewer for the comment. We have added conclusions at the end of the discussion, as follows:
“To conclude, the therapeutic landscape of LARC is becoming increasingly complex following the introduction of TNT along with the possibilities of escalation and de-escalation treatments. These strategies aimed at maximizing local and distant control while also preserving patients' quality of life. Therefore, the ideal and comprehensive LARC management should be based on both standard and “new” clinical-radiological parameters, that allow for accurate risk assessment, as well as consideration of patient’s preferences. All these evaluations must be enriched and implemented during the mul-tidisciplinary discussion, which must be considered indispensable. However, the large volume of data from recent trials has made outlining a precise treatment algorithm to guide clinicians in daily practice even more challenging and almost prohibitive. Hence, the therapeutic strategy should be the result of a full-shared decision between doctors and each patient. The definition and implementation of biomarkers remains an unmet need in this field. ctDNA may potentially enter clinical practice as a guide for enhancing neo-adjuvant therapy or surgery in the case of persistent positivity, or to open the possibility of less intensive approaches in the case of negativization.”
2) Chapter 8 is not scientifically clear and appears too long and should be anticipated before discussion and conclusion. Talking about the usefulness or otherwise of the liquid biopsy does not seem necessary, it is not essential for the topic treated. Maybe you propose it in another article.
2) We thank the reviewer for her/his suggestion: we have anticipated chapter 8 before discussion and we have summarized it. We strongly think that ctDNA would enter clinical practice, since in colorectal cancer is proving to be the strongest prognostic factor we ever had and possibly a predictive factor of chemotherapy efficacy. Therefore, we cannot overlook including it among the factors to be taken into consideration, since in the next future it will likely help us in patient selection, aiming to increase cure rates while reducing unnecessary toxicity.
3) Why Interpretation and takeaway is underlined? take off pls.
3) It was underlined to focus the attention of the reader on the key message of each paragraph. However, as you suggested, we have removed it.
Reviewer 3 Report
Comments and Suggestions for Authors
Congratulations on a well structured paper on what is becoming a very complex issue.
I believe the manuscript could be further improved by expanding on follow up criteria in all scenarios and elaborating the role of the endoscopic evaluation/criteria as well after the neoadjuvant protocols.
Figure 1. should be removed as it does not add value to the article.
With this minor revisions, I believe the article should be accepted.
Author Response
Reviewer 3:
1) Congratulations on a well structured paper on what is becoming a very complex issue. I believe the manuscript could be further improved by expanding on follow up criteria in all scenarios and elaborating the role of the endoscopic evaluation/criteria as well after the neoadjuvant protocols.
1) We thank the reviewer for having grasped the aim of our work and for the suggestion about expanding the follow up criteria. We have added a sub-paragraph in the NOM part, as follows:
4.1 The management of the follow up.
Unlike other cancer types, where follow-up protocols have long been standardized, locally advanced rectal cancer requires a personalized FU plan based on the chosen treatment approach. While no significant changes have been observed in the management of patients undergoing TNT followed by surgery (26) the NOM option needs a carefully tailored FU strategy to ensure the early detection and treatment of LR, given the high likelihood of achieving a cure. As we have already discussed, around 1 out of 3 patients developed a LR after NOM; nevertheless, long-term outcomes seem to be preserved if salvage surgery is promptly performed (31); it follows that a timely identification of a LR is mandatory. Furthermore, LR may anticipate distant recurrences, although this risk cannot be precisely determined (47). Consequently, regular systemic surveillance is mandatory. Unfortunately, due to the shades of uncertainness and the lack of randomized trials, no worldwide consensus on the best FU approach after NOM is still reached. Among the available data, most LRs occur within the first two years after the completion of treatment. Therefore, there is a general agreement to intensify FU in the first 2-3 years (55). National and international guidelines agree on the need to reassess patients locally every 3 months through clinical examination, digital rectal examination, and measurement of the tumor marker CEA. For pelvic MRI, the recommended interval ranges from 3 to 6 months, while reassessment with CT scans should be performed every 6 months. Table 3 presents and compares the most frequently adopted FU strategies to date. Patients who are offered W&W strategy should be fully aware of the necessity of a complete adherence to such a strict FU program, and clinicians should be able to assure availability of timely exams and to raise concerns about NOM in non-compliant patients. Surely, a better understanding of the disease and the identification of prognostic factors allowing patients’ stratification is highly needed, and both clinical and translational biomarkers are awaited.
|
Exams |
1st year |
2nd year |
3rd year |
4th-5th year |
||||||||
|
NCCN |
Dutch |
Brasil |
NCCN |
Dutch |
Brasil |
NCCN |
Dutch |
Brasil |
NCCN |
Dutch |
Brasil |
|
|
CE; RE |
Q3M |
Q3M |
Q2M |
Q3M |
Q6M |
Q2M |
Q6M |
Q6-12M |
Q2M |
Q6M |
Q6-12M |
Q6M |
|
CEA |
Q3-6M |
/ |
/ |
Q3-6M |
/ |
/ |
Q6M |
/ |
/ |
Q6M |
/ |
/ |
|
Proct. |
Q3M |
Q3M |
Q2M |
Q3M |
Q6M |
Q2M |
Q6M |
Q6-12M |
Q2M |
Q6M |
Q6-12M |
Q6M |
|
MRI |
Q6M |
Q3M |
Q3M |
Q6M |
Q6M |
Q3M |
Q6M |
Q6-12M |
Q3M |
/ |
Q6-12M |
Q6M |
|
CT scan |
Q6-12M |
? |
Q6M |
Q6-12M |
? |
Q6M |
Q6-12M |
Q12M |
Q6M |
Q6-12M |
Q12M |
Q6M |
Table 3: Surveillance programs according to NCCN guidelines (Version 4.2024), the Dutch Watch-and-Wait Consortium(56) and guidelines, and the Brazilians indications (57). Acronyms: CE= clinical examination; RE= rectal examination; CEA= carcinoembryonic antigen; Proct= proctoscopy; CEA= carcinoembryonic antigen; CT = computed tomography; MRI= Magnetic Resonance Imaging; QxM= every x month.
2) Figure 1. should be removed as it does not add value to the article. With this minor revisions, I believe the article should be accepted.
2) We thank the reviewer for her/his suggestion. However, we’d prefer to keep Figure 1 because it simply and graphically summarized all the different approaches, and could help the readers, especially if they are not in the field of rectal cancer, to have a rapid and general overview of the therapeutic scenario.
Round 2
Reviewer 2 Report
Comments and Suggestions for Authors
The paper appears improved and more logical in its development, could be published after a review by a native English speaker